# Mitigating Spurious Correlations in Patch-wise Tumor Classification on High-Resolution Multimodal Images

## Abstract

Patch-wise multi-label classification provides an efficient alternative to full pixel-wise segmentation on high-resolution images, particularly when the objective is to determine the presence or absence of target objects within a patch rather than their precise spatial extent. This formulation substantially reduces annotation cost, simplifies training, and allows flexible patch sizing aligned with the desired level of decision granularity. In this work, we focus on a special case, patch-wise binary classification, applied to the detection of a single class of interest (tumor) on high-resolution multimodal nonlinear microscopy images. We show that, although this simplified formulation enables efficient model development, it can introduce *spurious correlations* between patch composition and labels: tumor patches tend to contain larger tissue regions, whereas non-tumor patches often consist mostly of background with small tissue areas. We further quantify the bias in model predictions caused by this spurious correlation, and propose to use a debiasing strategy to mitigate its effect. Specifically, we apply GERNE, a debiasing method that can be adapted to maximize worst-group accuracy (WGA). Our results show an improvement in WGA by approximately 7% compared to ERM for two different thresholds used to binarize the spurious feature. This enhancement boosts model performance on critical minority cases, such as tumor patches with small tissues and non-tumor patches with large tissues, and underscores the importance of spurious correlation-aware learning in patch-wise classification problems.

## 1 Introduction

High-resolution imaging across diverse domains, such as biomedical microscopy, materials inspection, and remote sensing, provides detailed spatial and spectral information that enables precise analysis of complex structures [16, 1, 34, 26, 32, 19] . However, the sheer size and dimensionality of such images make direct processing computationally impractical, particularly for dense pixel-wise segmentation tasks [18, 37, 4, 38, 15, 12, 13, 2]. To address these challenges, large images are typically divided into smaller, fixed-size patches that can be efficiently processed by neural networks [27, 10, 11, 21, 14].

In many applications, the objective is not to achieve complete segmentation of each patch but rather to detect the presence or absence of specific, diagnostically or functionally significant class (e.g., tumors within tissue, defects in materials, or anomalies in spectral maps) [20, 22, 9, 29, 30]. In such cases, the problem can instead be reformulated as a patch-wise binary classification task [31, 25, 35, 7], where each patch is labeled positive if it contains at least one pixel of the target class, and negative otherwise, instead of pixel-wise labeling. This formulation reduces annotation effort and allows practitioners to balance spatial resolution, computational cost, and the minimal decision unit required for practical action [8, 24, 33]. For example, in medical imaging, this could correspond to the smallest tissue region a surgeon would resect.

Submitted to 39th Conference on Neural Information Processing Systems (NeurIPS 2025). Do not distribute.

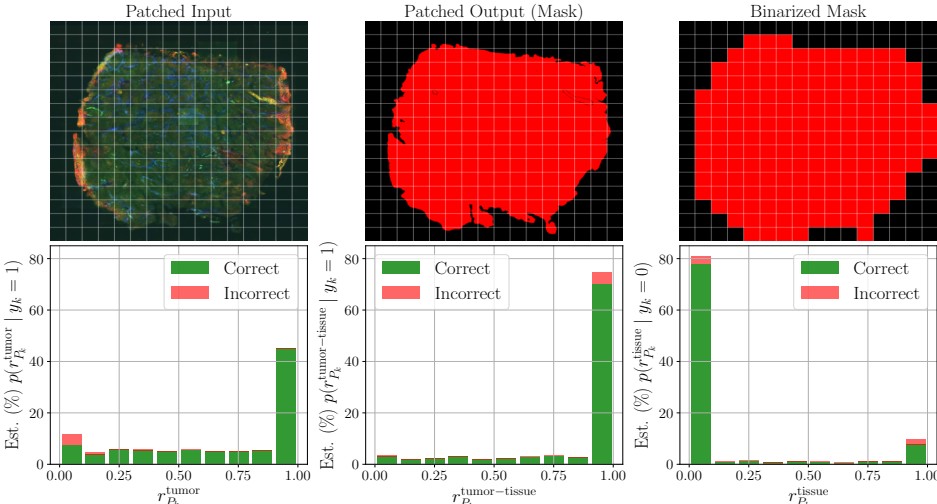

Figure 1: Illustration of the patch-wise classification pipeline and the emergence of spurious correlations between patch composition and binary labels. **Top row:** A high-resolution multimodal nonlinear image (left) sourced from [6] is divided into fixed-size patches to fit the neural network input. The corresponding pixel-wise segmentation mask is shown in the center (red: tumor tissue; black: non-tumor regions, including healthy tissue and background). The binarized patch-wise labels are shown on the right, where each patch is assigned a positive label (red) if it contains at least one tumor pixel, and a negative label (black) otherwise. **Bottom row:** Histogram-based analysis (on the test set of the dataset under study [6]) of patch composition reveals strong correlations between tissue size and patch labels. The estimated conditional distributions of $p(r_{P_k}^{\text{tumor}} \mid y_k = 1)$, $p(r_{P_k}^{\text{tumor-tissue}} \mid y_k = 1)$, and $p(r_{P_k}^{\text{tissue}} \mid y_k = 0)$ are shown from left to right as normalized histograms. Tumor patches tend to contain larger tissue regions, whereas non-tumor patches are dominated by background with minimal tissue. These correlations may introduce shortcuts for ERM-trained models, causing models to rely on non-causal features such as overall tissue size. To verify whether these spurious correlations are exploited by the model, we trained a standard ERM-based binary classifier and overlaid, for each distribution, the proportion of correct (green) and incorrect (red) predictions. The resulting pattern confirms that models can rely on non-causal cues such as overall tissue size, underscoring the need for debiasing strategies in patch-based learning frameworks.

While patch-wise classification simplifies model training and annotation, we note an important caveat: this simplification may inadvertently introduce *spurious correlations* [36, 17] between patch composition and class labels. In this work, we study patch-wise binary tumor classification on a high-resolution multimodal nonlinear image dataset [5]. We observe that tumor patches tend to contain larger tissue regions, while non-tumor patches are dominated by background. Such correlations can lead models to rely on non-causal cues (i.e., overall tissue size in our case), degrading performance on underrepresented groups where such cues are absent (e.g., accuracy on tumor patches with small tissue areas). To mitigate this, we apply GERNE [3], a gradient extrapolation-based debiasing method after binarizing the spurious feature (i.e., tissue size). Targeted debiasing improves model robustness, as measured by worst-group accuracy (WGA) [28], yielding more reliable and fair patch-wise image classification. Figure 1 illustrates the adopted patch-wise binary classification pipeline, depicts the distributions of relevant tissue ratios, highlighting the correlation between the tissue size and patch labels. It also shows correct and incorrect patch predictions of an ERM-trained model, demonstrating how these correlations bias the model and hinder generalization.

## 2   Problem Setup and Dataset

### 2.1   From High-Resolution Segmentation to Patch-wise Classification

Let $I \in \mathbb{R}^{H \times W \times M}$ denote a high-resolution image with height $H$, width $W$, and $M$ channels (e.g., multimodal channels). Processing the full image at once with standard convolutional or transformer-

based networks is often computationally infeasible due to memory and runtime constraints. A common strategy is to partition $I$ into $N$ smaller, fixed-size patches $\{P_k\}_{k=1}^N$, such that $I = \bigcup_{k=1}^N P_k$, where each patch $P_k \in \mathbb{R}^{h \times w \times M}$ has height $h \ll H$ and width $w \ll W$. This patch-based formulation reduces the computational burden while maintaining sufficient local spatial context for learning. The top-left panel in Figure 1 provides an example of this patching.

**Patch-wise Multi-Label Classification**   Instead of performing dense pixel-wise segmentation, the task can be reformulated as *patch-wise multi-label classification*, which is sufficient when the goal is merely to determine the presence or absence of each class within a patch, rather than delineating their precise spatial boundaries. Let there be $C$ classes of interest, and let the pixel-wise annotation for patch $P_k$ be denoted as $Y^k \in \{0,1\}^{h \times w \times C}$, where

$$Y_{i,j,c}^k = \begin{cases} 1 & \text{if pixel } (i,j) \text{ in patch } P_k \text{ belongs to class } c \\ 0 & \text{otherwise} \end{cases}, \quad c \in \{1, \ldots, C\}. \tag{1}$$

The corresponding patch-level label vector $\boldsymbol{y}_k \in \{0,1\}^C$ is then defined as:

$$y_{k,c} = \mathbf{1}\Big( \sum_{i=1}^h \sum_{j=1}^w Y_{i,j,c}^k \geq 1 \Big), \quad \forall c \in \{1, \ldots, C\}, \tag{2}$$

where $y_{k,c} = 1$ if class $c$ is present in patch $P_k$, and $0$ otherwise. This reformulation is particularly suitable when fine-grained segmentation is unnecessary, as it reduces annotation effort and computational cost while retaining essential semantic information about class occurrence.

**Special Case: Patch-wise Binary Classification**   In many applications, only the presence of a single class is of interest (e.g., tumor tissue in histopathology, defects in materials). In this case, the multi-label representation reduces to a single binary label per patch:

$$y_k = \mathbf{1}\Big( \sum_{i=1}^h \sum_{j=1}^w Y_{i,j,c'}^k \geq 1 \Big), \tag{3}$$

where $Y_{i,j,c'}^k \in \{0,1\}$ is the pixel-level label for the class of interest $c'$ (the target class); That is, a patch is labeled positive if it contains at least one pixel of the target class (tumor in our study), and negative otherwise.

## 2.2   Dataset under Study

We perform our study on a high-resolution multimodal nonlinear image dataset [5, 6], which provides paired images and pixel-wise segmentation masks. Each image $I \in \mathbb{R}^{H \times W \times M}$ contains multiple tissue types, including tumor and healthy regions, annotated at the pixel level. Each mutlimodal image has dimensions exceeding several thousand pixels per side, and has three different nonlinear imaging modalities ($M = 3$), capturing complementary structural and biochemical features. We partition the dataset into training, validation, and test sets following the same splits as in the original study [5]. We follow the labeling process described in Equation (3) to generate binary labels for each patch, considering the tumor label as the positive class whereas treating healthy tissue and background (referred to as "tissue to preserve" and "background" in [5]) as the negative class. Tissue regions can be inferred even when explicit labels are unavailable, as non-tissue (background) pixels are predominantly black. As illustrated in the top row of Figure 1, the left panel shows a multimodal input image, the center panel shows its corresponding pixel-wise binary mask for the tumor class, and the right panel shows the binarized patch-level labels.

# 3   Methodology

## 3.1   Spurious Correlations in Patch Composition

While patch-wise labeling simplifies model training and annotation, we observed that this formulation can introduce *spurious correlations* between patch composition and labels. To show this, we define different patch-level ratios based on the pixel-level labels $Y^k$ (see Equation 1), for $c \in$ {tumor tissue, healthy tissue, background} ($|P_k| = h \cdot w$):

**Tumor ratio** $r_{P_k}^{\text{tumor}}$ as the fraction of tumor pixels in a patch: $r_{P_k}^{\text{tumor}} = \frac{\sum_{i,j} Y_{i,j,\text{tumor tissue}}^k}{|P_k|}$.

**Tumor-to-Tissue ratio** $r_{P_k}^{\text{tumor-tissue}}$ as the fraction of tumor pixels relative to all tissue pixels:

$r_{P_k}^{\text{tumor-tissue}} = \frac{\sum_{i,j} Y_{i,j,\text{tumor tissue}}^k}{S_{P_k}}$, where $S_{P_k} = \sum_{i,j} \left( Y_{i,j,\text{tumor tissue}}^k + Y_{i,j,\text{healthy tissue}}^k \right)$.

**Tissue ratio** $r_{P_k}^{\text{tissue}}$ as the fraction of tissue pixels in a patch: $r_{P_k}^{\text{tissue}} = \frac{S_{P_k}}{|P_k|}$.

In the bottom row of Figure 1, we show the following *conditional distributions* of these patch ratios, approximated by normalized histograms computed over the test set :

$$p\left(r_{P_k}^{\text{tumor}} \,\middle|\, y_k = 1\right), \quad p\left(r_{P_k}^{\text{tumor-tissue}} \,\middle|\, y_k = 1\right), \quad p\left(r_{P_k}^{\text{tissue}} \,\middle|\, y_k = 0\right). \tag{4}$$

From this analysis, we observe that **tumor (positive)** patches tend to contain larger tissue regions, and these tissue regions are dominated by tumor tissue (with little or no healthy tissue), whereas **non-tumor (negative)** patches mostly consist of background with little or no tissue. This reveals a spurious correlation between the tissue size and the label. One possible cause for this correlation is the spatial geometry of the tissue and the binary labeling scheme. In the dataset under study, tumor regions tend to be convex-like structures and cover large portion of the high-resolution images, with little surrounding healthy tissues.

ERM-trained models often exploit spurious correlations in datasets when they are prevalent and easier to learn than the causal features [23]. To examine whether models exploit the spurious feature in our case, we visualize in Figure 1 (bottom row) the fraction of correctly and incorrectly classified patches along with each of the estimated distributions in Equation (4) (training setup described in Section 4). We observe that the model performs relatively better on patches where the tumor occupies a large fraction of the patch. Specifically, the proportion of correctly classified positive samples, relative to the total number of samples within the corresponding histogram bins, is higher for patches with large tumor coverage (high $r_{P_k}^{\text{tumor}}$) compared to those with small tumor coverage. The opposite trend is observed for non-tumor patches, where bins that represent high tissue ratios exhibit a relatively higher proportion of misclassifications. This indicates that ERM-trained models can exploit correlations between tissue size and label rather than relying solely on tumor-specific features, underscoring the need for debiasing strategies to improve performance on underrepresented patch types.

### 3.2 Debiasing Spurious Correlations in Patch-wise Classification

Bias mitigation has been extensively studied in computer vision, particularly in contexts where models inadvertently rely on spurious, non-causal features such as background or texture cues. Extensive prior work has considered discrete spurious attributes (e.g., gender, color, or environment), which enable group-based analysis and optimization [36]. To leverage these established methods, we discretize the tissue-size attribute into two categories, small and large (Section 3.2.1). We then apply a recently proposed gradient extrapolation-based debiasing technique, GERNE [3] (Section 3.2.2), which can be adapted to maximize WGA through tuning its hyperparameters.

### 3.2.1 Binarization of Tissue-Size Spurious Feature

we binarize $r_{P_k}^{\text{tissue}}$ using a threshold $\tau \in [0, 1]$, creating a binary spurious attribute $z_{P_k}$ defined as:

$$z_{P_k} = \begin{cases} 0, & \text{if } r_{P_k}^{\text{tissue}} < \tau \quad \text{(small tissue)}, \\ 1, & \text{if } r_{P_k}^{\text{tissue}} \geq \tau \quad \text{(large tissue)}. \end{cases} \tag{5}$$

As a result, we obtain four distinct groups corresponding to the combinations of the binary patch label and the binary spurious attribute.

### 3.2.2 Gradient Extrapolation for Debiased Learning (GERNE)

In this work, we adopt GERNE [3] to reduce the negative effect of spurious correlations, owing to its conceptual simplicity, ability to maximizing for WGA, and computational efficiency.

Before each model parameters update, GERNE constructs two mini-batches with different levels of spurious correlations: a *biased* batch $B_b$, reflecting the dataset's inherent bias, and a *less-biased*

137 batch $B_{lb}$ with a more blaanced group distribution. Let $\mathcal{L}_b$ and $\mathcal{L}_{lb}$ denotes the corresponding training
138 losses, and their gradients with respect to the model parameters $\theta$ be $g_b = \nabla_\theta \mathcal{L}_b$ and $g_{lb} = \nabla_\theta \mathcal{L}_{lb}$.
139 GERNE computes an *extrapolated gradient*: $g_{\text{ext}} = g_{lb} + \beta \left( g_{lb} - g_b \right)$, which is then used to update
140 the model parameters. The extrapolation factor $\beta$ is a hyperparameter that controls the degree of
141 directing the learning process in a debiasing direction.

## 4   Experimental Results

**Dataset and Implementation Details.**   We generate training, validation, and test sets following
the preprocessing pipeline in [5]. We assign patch-level binary labels to the extracted patches using
the procedure described in Section 2.1 (see Figure 1 (top row)). We employ a ResNet-50 backbone
pretrained on ImageNet-1K, followed by one fully layer with two outputs, and use Cross Entropy as
loss function to minimize.

**Evaluation Metric.**   We use the validation accuracy for model selection and hyperparameter tuning,
and report the following two metrics on the test set for each experiment: WGA, which measures the
lowest accuracy across groups, and Balanced-Class Accuracy (BCA), which computes the average
accuracy across classes. For the ERM-trained model, we use both metrics as evaluation criteria,
whereas for GERNE, we use only WGA.

**Results.**   In  Table 1, we report the results for two different thresholds $\tau = 0.1$ and $\tau = 0.03$.

Table 1: Comparison of models performance using ERM and GERNE under two different tissue ratio
thresholds. Accuracies (%) are reported using two metrics on test set: WGA, and BCA. Results are
averaged over three trials (mean $\pm$ std). Best results are **bolded**.

| Method | Eval. Metric | $\tau = 0.1$ | | $\tau = 0.03$ | |
|---|---|---|---|---|---|
| | | WGA | BCA | WGA | BCA |
| ERM | BCA | 63.75 $\pm$0.68 | **93.51$\pm$0.51** | 48.94$\pm$1.32 | **93.47$\pm$0.58** |
| ERM | WGA | 73.54 $\pm$0.82 | 92.30$\pm$0.29 | 69.46 $\pm$0.91 | 92.94$\pm$0.66 |
| GERNE | WGA | **80.93$\pm$0.94** | 90.90$\pm$0.22 | **76.40$\pm$0.46** | 88.63$\pm$0.57 |

154 In table 1, we observe two key findings. First, for ERM, WGA on test set is higher when this metric
155 is used as the evaluation metric compared to using BCA, supporting the conclusion in [36] about the
156 importance of carefully selecting the evaluation metric. Second, GERNE consistently outperforms
157 ERM in terms of WGA for the same evaluation metric (WGA) across both tissue ratio thresholds,
158 improving WGA by approximately 7% in both cases, with only a minor decrease in BCA. The
159 improvement is particularly important for small-tissue tumor patches (minority group) that include
160 tumor boundary cases, where detecting the presence or absence of tumor is especially challenging.
161 Accurate classification of these patches is critical for surgical decision-making, highlighting the
162 clinical relevance of mitigating spurious correlations in patch-wise tumor classification.

## 5   Conclusion

In this work, we investigate the emergence and mitigation of spurious correlations in patch-wise
binary tumor classification on high-resolution multimodal imaging data. Through systematic analysis,
we show that tissue size acts as a spurious attribute strongly correlated with patch labels. To counter
the negative effects of this spurious correlation, we formulate the spurious attribute as a binary
variable and apply a gradient extrapolation-based debiasing method. We demonstrate consistent
improvements in WGA over ERM across two different thresholds, which is particularly important for
tumor patches with small tissue regions, cases that are challenging for surgeons to assess.

Future work will examine how patch size affects spurious correlations and model performance, and
explore alternative debiasing strategies that address multiple or continuous spurious attributes without
relying on binarization. In addition, we aim to investigate how such correlations arise in and affect
the full segmentation tasks, and extend our analysis to other high-resolution domains, such as remote
sensing and materials science, where patch-wise classification may similarly introduce bias.

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
