# OpenReview forum: "Mitigating Spurious Correlations in Patch-wise Tumor Classification on High-Resolution Multimodal Images"
_EurIPS.cc/2025/Workshop/UPLB — UPLB2025_

### Official Review · Reviewer_zBSt · 2025-10-27
**Mitingation in Tumor classification**

**Rating:** 5
**Confidence:** 3

**Review:**

This work focus on the tumour classification on large images, where the classification is done on small patches (mainly for computational reason). The paper study a class of images where the small patches are classified to contain tumour (or not) while trying to counterbalance the effect of spurious correlations.
In this work, the main spurious correlation considered it the fact that a path tends to be classified as "positive" (or containing tumour) where it contains a high amount of tissue cells. The authors used a "debiaising" method based on a gradient extrapolation that seems to perform much better than a more naive approach.

I'm very puzzled by the dataset considered and the spurious effect that is discussed. The dataset itself is constructed based on this spurious correlation: each patch that contains at leach one tumour pixel is then classified as "tumorous". It thus seems to me that the authors intend to avoid an effect that they put themselves in the dataset. Another comment: at that point, I do not understand why a more precise method can not be applied on the patches, such as pixel-segmentation, to avoid having to classify an entire patch as tumorous.

---

### Decision · Program_Chairs · 2025-11-03

Accept (Poster)